# Influence of Salt on the Self-Organization in Solutions of Star-Shaped Poly-2-alkyl-2-oxazoline and Poly-2-alkyl-2-oxazine on Heating

**DOI:** 10.3390/polym13071152

**Published:** 2021-04-04

**Authors:** Tatyana Kirila, Anna Smirnova, Alla Razina, Andrey Tenkovtsev, Alexander Filippov

**Affiliations:** Institute of Macromolecular Compounds of the Russian Academy of Sciences, Bolshoy Pr. 31, 199004 Saint Petersburg, Russia; av.smirnova536@gmail.com (A.S.); allarazina@yahoo.com (A.R.); avt@hq.macro.ru (A.T.); afil@imc.macro.ru (A.F.)

**Keywords:** star-shaped polymer, poly-2-alkyl-2-oxazines and poly-2-alkyl-2-oxazolines, saline solution, thermoresponsive polymers

## Abstract

The water–salt solutions of star-shaped six-arm poly-2-alkyl-2-oxazines and poly-2-alkyl-2-oxazolines were studied by light scattering and turbidimetry. The core was hexaaza[2_6_]orthoparacyclophane and the arms were poly-2-ethyl-2-oxazine, poly-2-isopropyl-2-oxazine, poly-2-ethyl-2-oxazoline, and poly-2-isopropyl-2-oxazoline. NaCl and *N*-methylpyridinium *p*-toluenesulfonate were used as salts. Their concentration varied from 0–0.154 M. On heating, a phase transition was observed in all studied solutions. It was found that the effect of salt on the thermosensitivity of the investigated stars depends on the structure of the salt and polymer and on the salt content in the solution. The phase separation temperature decreased with an increase in the hydrophobicity of the polymers, which is caused by both a growth of the side radical size and an elongation of the monomer unit. For NaCl solutions, the phase separation temperature monotonically decreased with growth of salt concentration. In solutions with methylpyridinium p-toluenesulfonate, the dependence of the phase separation temperature on the salt concentration was non-monotonic with minimum at salt concentration corresponding to one salt molecule per one arm of a polymer star. Poly-2-alkyl-2-oxazine and poly-2-alkyl-2-oxazoline stars with a hexaaza[2_6_]orthoparacyclophane core are more sensitive to the presence of salt in solution than the similar stars with a calix[n]arene branching center.

## 1. Introduction

The key property of “smart polymers”, which determines a wide range of their practical use, is a nonlinear response to an external signal. In the case of thermoresponsive polymers, the phase transition in aqueous solutions is induced by temperature change. Accordingly, the temperature variation is a simple way to control the behavior of their solutions [1,2,3,4,5,6]. Thermoresponsive polymers are highly appealing for medical applications and biotechnology if the phase separation temperature is close to body temperature [7,8,9,10,11,12,13,14]. Polymers used in biomedical applications must be non-toxic, biocompatible, stable in biological media, biodegradable, and/or completely excreted from the body. These requirements are satisfied by poly-2-alkyl-2-oxazolines (PAlOx) and poly-2-alkyl-2-oxazines (PAlOz), many of which exhibit a thermosensitivity with a lower critical solution temperature.

PAlOx and PAlOz, sometimes called pseudo-polypeptoids, are obtained by (living) cationic ring-opening polymerization of corresponding monomers [15,16,17,18,19,20,21]. In recent years, a large number of thermoresponsive PAlOx of various chemical structures have been obtained, including statistical, block and gradient copolymers, and graft and star-shaped polymers [22,23,24,25]. The processes of polymerization, in particular, the kinetics of polymerization upon initiation by alkyl halides, tosylates, nosylates, and triflates have been studied in detail [26,27]. Regularities of behavior in aqueous solutions with temperature variation were established for PAlOx, and features that distinguish them from other thermosensitive polymers were revealed [28,29,30,31,32,33,34]. It has been found that the introduction of fragments ready for effective binding of inorganic ions and organic compounds, for example, azamacrocycles, is an effective strategy for the construction of supramolecular structures, thereby allowing to simulate the processes occurring in living nature [35,36,37].

The polymerization process of PAlOz is characterized by low polymerization rate constants and a high chain transfer rate, which makes it difficult to obtain high molar mass samples [19,21]. Until recently, this was one of the reasons for the small number of works devoted to the study of this promising class of thermosensitive polymers. Nonetheless, at the present time, the effect of the structure of the monomer unit and end groups on the properties of linear PAlOz [14,19,21,38,39,40], including their thermosensitivity [14,41,42], has been established. It was shown that the hydration of PAlOz is determined by the structure of the side radical. An additional methylene group in the backbone makes PAlOz more hydrophobic than PAlOx with the same side alkyl radical, which leads to a decrease in the cloud point of aqueous solutions [42]. The most significant is that an increase in the binding of water-insoluble drugs was found for PAlOz in comparison with PAlOx [14,41,43]. The latter indicates good prospects for the use of PAlOz and their copolymers in medicine.

It is well known that star-shaped polymers have very good prospects for use in medical applications [44,45], for example, for drug delivery [46], for selective adhesion of cancer cells [47], in tissue engineering, and cell uptake [48,49]. This circumstance has intensified research in the field of synthesis and determination of the properties of stimuli-sensitive polymers with complex architecture [50,51], including star-shaped PAlOx. The studies performed have revealed a number of interesting regularities in the behavior of polymers with complex architecture in aqueous solutions. For example, for a number of classes of star-shaped polymers, the influence of the core structure, the length and number of arms on the self-organization and aggregation has been established [12,52,53,54,55,56,57]. Recently, for the first time, star-shaped PAlOz were synthesized and studied [58,59,60].

The use of thermoresponsive PAlOx and PAlOz as materials for drug delivery is due to their ability to form intra- and intermolecular hydrogen bonds, resulting in the compaction and aggregation of individual polymer chains [61,62,63,64,65]. The physiological media is a complex system containing a variety of ions that affect the balance of hydrogen bonds. Accordingly, the behavior of thermoresponsive polymers in aqueous solutions and physiological media can differ significantly. For example, even a small NaCl content in the solution significantly changed the phase separation temperatures [66,67,68]. Moreover, the presence of salt affects the thermosensitivity of the linear and star-shaped polymers in different ways [60,61,62,63,64,65,66,67,68,69].

The goal of this work is to analyze the effect of the chemical structure and concentration of low molecular weight salts on the behavior of the star-shaped PAlOx and PAlOz in water–salt solutions. Four star-shaped polymers with a hexaase[2_6_]orthoparacyclophane (CPh6) core were investigated, namely, poly-2-ethyl-2-oxazine (CPh6-PEtOz), poly-2-isopropyl-2-oxazine (CPh6-PiPrOz), poly-2-ethyl-2-oxazoline (CPh6-PEtOx), and poly-2-isopropyl-2-oxazoline (CPh6-PiPrOx) (Figure 1). NaCl and *N*-methylpyridinium *p*-toluenesulfonate (*N*-PTS, Figure 2) were used as salts. *N*-PTS can be considered as a model for cetylpyridinium chloride known for its antimicrobial and antifungal effects. *N*-PTS influences the self-organization in solution of thermoresponsive polymers [70], because both *N*-methyl pyridinium cation and tosylate anion affect the hydrogen bond network of water [71,72,73,74].

## 2. Materials and Methods 

### 2.1. Polymer Star Synthesis

The synthesis and characterization of star-shaped six-arm thermoresponsive poly-2-alkyl-2-oxazolines (CPh6-PAlOx) and poly-2-alkyl-2-oxazines (CPh6-PAlOz) with a hexaase[2_6_]orthoparacyclophane core has been described in detail previously [59]. CPh6-PAlOx and CPh6-PAlOz were synthesized by cationic ring-opening polymerization of the corresponding 2-alkyl-2-oxazoline or 2-alkyl-2-oxazine derivative. The molar masses (MM) and hydrodynamic characteristics of the samples were determined in chloroform dilute solutions using the sedimentation-diffusion analysis and viscosity. MM were moderate: 23,000 g⋅mol^−1^ for CPh6-PEtOz, 20,000 g⋅mol^−1^ for CPh6-PiPrOz, 15,000 g⋅mol^−1^ for CPh6-PEtOx, and 14,000 g⋅mol^−1^ for CPh6-PiPrOx [59]. Accordingly, the molar masses of CPh6-PAlOz are slightly higher than MM of CPh6-PAlOx.

The solvents and reagents (all Sigma Aldrich, St. Louis, MO, USA) were purified and dried according to the standard techniques. Trianglamine (1) [11] as well as 2-alkyl-2-oxazolines and 2-alkyl-2-oxazines [41] were synthesized by the generally applied methods.

### 2.2. Solution Investigation

The behavior of CPh6-PAlOz and CPh6-PAlOx in water–salt solutions was studied at polymer concentration *c* = 0.0050 g⋅cm^−3^. For NaCl solutions, the salt concentrations *c*_salt_ were selected as one NaCl formula unit per one macromolecule, per one arm of the polymer star and per one monomer unit. Besides, physiological saline (0.154 M) and pure aqueous solutions were investigated. In the case *N*-PTS solutions, the *c*_salt_ values were chosen in a similar way: one *N*-PTS molecule per one macromolecule, per one arm of the polymer star, and per one monomer unit. To expand the range of *N*-PTS content, solutions at *c*_salt_ ≈ 0.10 and 0.15 M were prepared and studied. Thus, for both water–salt solvents, the salt concentration varied from 0–1.54 M.

The solutions and solvent were filtered into cells previously dedusted by benzene. Chromafil Xtra filters (Macherey-Nagel, Dueren, Germany) with a PTFE membrane with the pore size of 0.45 μm were used.

The self-organization in water–salt solutions of CPh6-PAlOz and CPh6-PAlOx was studied by light scattering and turbidimetry on a PhotoCor Complex setup (Photocor Instruments Inc., Moscow, Russia) with a sensor for measuring optical transmission. The light source was the Photocor-DL diode laser with wavelength λ = 659.1 nm and controllable power up to 30 mW. The correlation function of the scattered light intensity was obtained using the Photocor-PC2 correlator with 288 channels and processed using the DynalS software. The solution temperature *T* was changed discretely within the interval from 9–80 °C, with the steps ranging from 0.5 (near phase separation) to 6 °C (low temperatures). The temperature was regulated with the precision of 0.1 °C. The heating rate was 1.5 °C·min^−1^.

The measurement procedure has been described in detail previously [55]. After the given temperature was achieved, all solution characteristics (light scattering intensity *I*, optical transmittance *I**, and hydrodynamic radii *R*_h_ of the scattering particles) began to change with time *t*. If intensity *I* changed at a high rate, the dependence of *I* (at the scattering angle 90°) and *I** on time was measured only. The analysis of these dependencies makes it possible to determine the time of establishment of the “equilibrium” state of the system, in which *I*, *I** and *R*_h_ cease to change in time at given temperature. The hydrodynamic radii Rh of dissolved particles were determined when the intensity changed very weakly or independent of *t*. It should be noted that the values of *R*_h_ can be obtained correctly if the light scattering intensity differs no more than 1% from its average value. Figure 3 as an example demonstrates the dependences of relative intensity *I*/*I*_max_ of scattered light on the hydrodynamic radius *R*_h_ of scattering species for CPh6-PEtOz water–salt solutions (*I*_max_ is the maximum value of light scattering intensity *I* for a given solution). It is necessary to emphasize that the experiment time was equal to 1800 s at least at each temperature even if the measured characteristics did not depend on time. In “equilibrium” conditions, the angle dependences of *I* and *R*_h_ were also analyzed within intervals from 45–135° in order to justify the diffusion process (Figure 4).

## 3. Results and Discussion

### 3.1. Behavior of Star-Shaped Six-Arm Pseudo-Polypeptoids in Water-Salt Solutions at Low Temperatures

The behavior of water–salt solutions of the studied polymer stars depends on the chemical structure of both the arms and the salts. In CPh6-PAlOx solutions, the addition of NaCl and N-PTS does not change the set of scattering objects. Figure 5 shows the dependences of the hydrodynamic radii *R*_h_ of the particles present in the solutions on the salt concentration *c*_salt_ for CPh6-PAlOx. For all values of *c*_salt_, two types of particles with radii *R*_m_ (small particles) and *R_s_* (large particles) were found in CPh6-PAlOx solutions. For both salts, *R*_m_ did not depend on the salt content. In the case of CPh6-PEtOx, the average values <*R*_m_> = (7.4 ± 0.4) nm for NaCl solutions and (7.0 ± 0.4) nm for *N*-PTS solutions are approximately 2.5× larger than the hydrodynamic radius *R*_h–D_ = 3.0 nm of CPh6-PEtOx molecules [59]. For the more hydrophobic CPh6-PiPrOx, the hydrodynamic radius <*R*_m_> is about 18 nm in both solvents, while the macromolecule radius *R*_h–D_ was 2.6 nm [59]. Consequently, just as in pure water in water-salt solutions of CPh6-PAlOx, the species with radius *R*_m_ are small aggregates, the reason for the formation of which is the interaction of hydrophobic CPh6 cores. These so-called micelle-like structures [75,76,77,78] were often observed in solutions of star-shaped PAlOx [79,80]. Taking into account that the form of star-shaped macromolecules with short arms and micelle-like structures [81] is close to spherical, the aggregation degree *m*_agg_ can be estimated by comparing the volumes of macromolecules and aggregates using the formula:*m*_agg_ ≈ (*R*_m_/*R*_h–D_)^3^(1)

Using Equation (1), it is also assumed that the densities of macromolecules and micelle-like structures are the same. For CPh6-PEtOx, the aggregation degree is low (*m*_agg_ ≈ 15), whereas for the more hydrophobic CPh6-PiPrOx, the *m*_agg_ value approaches 300.

As for large scattering objects with a hydrodynamic radius *R*_s_, these are large loose aggregates. The addition of NaCl and *N*-PTS to the solution has a different effect on the size of these aggregates. The value *R*_s_ is independent of the concentration of *N*-PTS and increases with growth of NaCl content in solution (Figure 5).

In the case of CPh6-PAlOz, the addition of salts leads to a change in the set of scattering objects (Figure 6). At low temperatures, two types of species were also observed in aqueous solutions of these star-shaped polymers. However, unlike CPh6-PAlOx, the smaller particles were macromolecules. Indeed, the hydrodynamic radius *R*_f_ of these objects coincided within the experimental error with the radius *R*_h–D_ of macromolecules [59]. In NaCl solutions of CPh6-PiPrOz, macromolecules and large aggregates were present in the studied range of the salt concentration *c*_salt_. The *R*_f_ value did not depend on the NaCl content, while the *R*_s_ radius increased with the growth of NaCl concentration. A completely different behavior was observed for CPh6-PiPrOz solutions in the presence of *N*-PTS. Macromolecules were detected only at low *N*-PTS content. At *c*_salt_ = 0.039 M, particles with a hydrodynamic radius *R*_m_ appeared in solutions, and at *c*_salt_ > 0.01 M, species with radius *R*_f_ were not observed by dynamic light scattering (Figure 6). The *R*_m_ values do not depend on the salt concentration. The average value <*R*_m_> = (6.1 ± 0.4) nm, and therefore, taking into account that *R*_h–D_ = 3.0 nm [59], the aggregation degree is *m*_agg_ ≈ 8. This is half the *m*_agg_ value for *N*-PTS solutions of CPh6-PEtOx. At a low *N*-PTS content, the hydrodynamic size of large aggregates is close to 70 nm, and at *c*_salt_ > 0.05 M, the *R*_s_ value increased, reaching 110 nm. Thus, at *c*_salt_ > 0.07 M in *N*-PTS solutions of CPh6-PiPrOz, micelle-like structures and large aggregates existed, which coincides with the set of scattering objects in water-salt solutions of CPh6-PAlOx.

In both water–salt solutions, the behavior of CPh6-PEtOz was similar to that observed for *N*-PTS solutions of CPh6-PiPrOz, namely, at a certain concentration *c*_salt_, micelle-like aggregates were formed in the solutions. Their hydrodynamic radius *R*_m_ did not change with *c*_salt_. The average values <*R*_m_> were (6.6 ± 0.4) nm and (6.9 ± 0.4) nm for solutions with NaCl and *N*-PTS, respectively. Small radii of micelle-like aggregates indicate that they contain a small number of macromolecules, and the estimation of the aggregation degree according to Equation (1) leads to a value of *m*_agg_ ≈ 7. Thus, the size of micelle-like aggregates formed in water–salt solutions of CPh6-PAlOz is smaller than the corresponding characteristics for CPh6-PAlOx. This can be explained by the fact that the arms of the CPh6-PAlOz molecules are longer [59] and better screen the nucleus. As regards the size of large aggregates, for CPh6-PEtOz, the *R*_s_ value does not depend on the *N*-PTS content and increases with the NaCl concentration (Figure 6).

Concluding the discussion of the behavior of water–salt solutions of CPh6-PAlOz at low temperatures, the following fact should be noted. The appearance of micelle-like aggregates in all cases occurs at a concentration *c*_salt_, which approximately corresponds to one salt molecule per one arm of a polymer star.

### 3.2. Temperature Dependences of Characteristics of Star-Shaped Pseudo-Dendrimers Water-Salt Solutions

All the results discussed below were obtained for a state of the investigated solutions when their characteristics (light scattering intensity *I*, optical transmission *I**, hydrodynamic radii of scattering species *R*_h_, etc.) do not change with time. For the systems under study, the establishment time *t*_eq_ of such “equilibrium” state after a discrete change in temperature is rather long. This is illustrated in Figure 7, which shows typical dependences of *I* and *I** on time *t*. The moment when the solution temperature reached a given value was taken as *t* = 0. For each solution, the *t*_eq_ value depended on temperature. As well as for other thermoresponsive polymers [82], at each *c*_salt_ value, the establishment time *t*_eq_ had a maximum value *t*_eq_^(max)^ near the onset of phase separation. For the studied polymer stars, no systematic change in *t*_eq_^(max)^ was found with a change in the salt content. Average values <*t*_eq_^(max)^> of maximum value of establishment time for each polymer are given in Table 1. They are noticeably less than <*t*_eq_^(max)^> for star-shaped eight- and four-arm PAlOx with a calix[n]arene core [82]. This is probably the effect of the structure of the branching center on the rate of self-organization of star-shaped polymers. The data in Table 1 show that the maximum establishment time for stars with side isopropyl groups is greater than <*t*_eq_^(max)^> for polymers with ethyl groups. Thus, an increase in the volume of the side radical slows down the aggregation processes in water–salt solutions of the star-shaped CPh6-PAlOz and CPh6-PAlOx. Note that earlier a decrease in the <*t*_eq_^(max)^> value with the passage from PiPrOx to PEtOx was found for PAlOx stars with a calix[n]arene core [81]. On the other hand, the influence of the salt structure on the times of establishing the equilibrium state has not been revealed. Indeed, for all studied polymers, the values of <*t*_eq_^(max)^> in solutions with NaCl and N-PTS coincide within the experimental error.

On heating, a structural-phase transition was observed in solutions of star-shaped CPh6-PAlOz and CPh6-PAlOx. Phase separation temperatures were measured by turbidimetry and light scattering methods (Figure 8). The temperature of its onset *T*_1_ was determined as the beginning of the sharp decrease in optical transmittance *I** and a rapid increase in the light scattering intensity *I*. At the temperature *T*_2_, which reflects the finishing of phase separation, the optical transmission becomes zero. At this temperature, for most of the studied solutions, the intensity *I* reaches maximum value. Note, that for the CPh6-PEtOx and CPh6-PEtOz solutions at low salt concentration (*c*_salt_ < 0.005 M), the temperature *T*_2_ could not be determined because it was too high (>85 °C).

As seen in Figure 8, with heating, the light scattering intensity began to change at relatively low temperatures. For example, for CPh6-PEtOx in NaCl solutions, a reliably measurable increase in *I* was observed at 45 °C. At this temperature, the *I* value exceeds the value of light scattering intensity *I* at 21 °C (*I*_21_) by 10%, i.e., *I*/*I*_21_ = 1.1. A further increase in *T* leads to a slow increase in the *I* value up to a temperature of onset of phase separation *T*_1_ (for discussed solution, 70 °C according to turbidimetry data), at which *I*/*I*_21_ = 3.3. Above *T*_1_, the rate of change in the light scattering intensity on heating increases by an order of magnitude. The dependence of *I* on *T* is caused by the increase in the size *R*_s_ of large aggregates on heating, while the values of *R*_f_ and *R*_m_ do not change with temperature. (Figure 9). The change in *R*_s_ is not high, but it is detected rather reliably. Therefore, at *T* < *T*_1_, the dominant process in the solutions of the studied star-shaped polymers was aggregation as a result of an increase in the dehydration degree of arms with temperature and the formation of intermolecular hydrogen bonds.

At *T* > *T*_1_, a sharp increase in the size of large aggregates was observed, and at the temperature of the phase separation finishing, the *R*_s_ values reached hundreds of nanometers and even microns. Above *T*_2_, the radii of large aggregates slightly decreased, which reflects the macromolecule compaction. Note that, in the studied temperature range, the sizes of micelle-like structures did not depend on temperature, and in the phase transition (near *T*_1_) these particles ceased to be detected by the dynamic light scattering. Therefore, they joined with large aggregates or formed new supramolecular structures.

### 3.3. Influence of Salt Content on Phase Separation Temperatures

Figure 10 shows the dependences of the phase separation temperature *T*_1_ on the salt concentration *c*_salt_ for water–salt solutions of CPh6-PAlOz and CPh6-PAlOx. It is clearly seen that the NaCl and *N*-PTS affect the behavior of the investigated star-shaped pseudo-polypeptoids in different ways. For NaCl solutions, an increase in *c*_salt_ leads to a monotonic decrease in *T*_1_, the rate of which decreases in the region of high values of *c*_salt_. Similar dependences were observed earlier for PAlOx of different architectures [69,83,84,85,86,87], as well as linear and star-shaped PEtOz [60]. In the case of N-PTS solutions, for all polymers, the phase separation temperatures decline very quickly in the region of low salt content. At a concentration *c*_salt_ corresponding to approximately one *N*-PTS molecule per one arm of polymer star, the decrease in *T*_1_ slows down, the phase separation temperature reaches a minimum value *T*_1_^(min)^, and then the *T*_1_ value begins to rise with increasing *N*-PTS content. Thus, for the studied polymer stars, at a low content in solution, *N*-PTS manifests itself as a kosmotropic agent, and at high *c*_salt_, *N*-PTS exerts chaotropic activity. What agent, chaotropic or kosmotropic, is a particular salt is a complex problem and its analysis, in particular the study of the interaction of thermoresponsive polymers with salt, has been devoted to a large number of works [66,67,71,72,88,89,90]. Analyzing the effect of salt on the behavior of polymer solution, it is necessary to take into account not only the chemical structure of the polymer and salt, but also their concentration in solution, ionic strength, temperature, etc.

The effect of NaCl and *N*-PTS on the behavior of CPh6-PAlOz and CPh6-PAlOx solutions depends on the arm structure. It is convenient to analyze the effect of the chemical structure of arms on the characteristics of water–salt solutions of the studied stars, comparing not only the dependences of the phase separation temperatures on the salt content (Figure 10), but also the dependences Δ*T*_1_ = *T*_1_^(0)^ − *T*_1_^(c)^ on *c*_salt_ (Figure 11), where *T*_1_^(0)^ is the temperature of onset of phase separation at *c*_salt_ = 0 and *T*_1_^(c)^ is this temperature at a given *c*_salt_. The Δ*T*_1_ value determines the change in the phase separation temperature upon salt addition. As can be seen in Figure 10, for both salts in the studied range of *c*_salt_, the phase separation temperatures decrease in the series CPh6-PEtOx–CPh6-PEtOz–CPh6-PiPrOx–CPh6-PiPrOz. Therefore, in water–salt solutions, a regularity, which is valid for solutions of CPh6-PAlOz and CPh6-PAlOx in water, is preserved. The Δ*T*_1_ values in NaCl solutions change in the same way (Figure 11). In solutions containing *N*-PTS, this sequence occurs only at low *c*_salt_ concentrations (Figure 11). At *c*_salt_ > 0.02 M, the Δ*T*_1_ value for CPh6-PEtOz becomes lower than the corresponding characteristic for CPh6-PiPrOx. This is due to the fact that after reaching the minimum value *T*_1_^(min)^, the phase separation temperature for stars containing side ethyl and isopropyl groups in the arms increases with different rates. For CPh6-PEtOz and CPh6-PEtOx, the temperature *T*_1_ at high *c*_salt_ exceeded the value *T*_1_^(min)^ by 29 and 18 °C, respectively. For CPh6-PiPrOx and CPh6-PiPrOz, the increase in *T*_1_ in the region of high *N*-PTS content was smoother, and the change in *T*_1_ did not exceed 4 °C (Figure 11).

As is known, with the same structure of side groups, PAlOz are more hydrophobic than PAlOx. Accordingly, at the given concentration and molar mass of the polymer, the phase separation temperatures for aqueous solutions of PAlOz are lower than for the corresponding PAlOx [14,43,91]. This regularity is observed for water–salt solutions of the studied star-shaped polymers, namely, at all salt concentrations *c*_salt_; in *N*-PTS and NaCl solutions, the temperature *T*_1_ decreased with passage from CPh6-PAlOx to CPh6-PAlOz. Note that the molar masses of the CPh6-PAlOz samples are higher than the MM of CPh6-PAlOx. An increase in the MM usually leads to a growth in the phase separation temperatures [85,92,93,94]. Consequently, some contribution to the observed difference in the *T*_1_ values for water–salt solutions star-shaped of CPh6-PAlOz and CPh6-PAlOx can be made by changing MM.

As seen in Figure 11, for the star-shaped CPh6-PEtOz, the maximum change in the phase separation temperature Δ*T*_1_ is approximately the same in both water–salt solvents: The maximum Δ*T*_1_ value is around 45 °C. For CPh6-PiPrOz, the maximum Δ*T*_1_ values are noticeably lower (Δ*T*_1_ ~ 10 °C), but they also coincide in different solvents. In the case of CPh6-PAlOx, a similar behavior was detected for CPh6-PiPrOx, while for CPh6-PEtOx, the maximum Δ*T*_1_ values in NaCl and *N*-PTS solutions differed by 10 °C.

Comparison of the obtained results with the literature data for other star-shaped pseudo-polypeptoids shows that their behavior in water–salt solutions depends on the core structure. For example, for water–salt solutions of eight-arm star-shaped poly-2-isopropyl-2-oxazoline with calix [8] arene core (at polymer concentration *c* = 0.0050 g⋅cm^−3^), the dependence of the phase separation temperature on the N-PTS content was monotonic [70], and the decrease in *T*_1_ in the *c*_salt_ range from 0–0.06 M was about 4 °C. Note that for the CPh6-PiPrOx studied in this work, the Δ*T*_1_ value exceeded 20 °C.

Figure 12 compares the dependences of Δ*T*_1_ on *c*_salt_ for the six-arm CPh6-PEtOz and CPh6-PiPrOx studied in this work, four-arm PEtOz with a calix [4] arene core (C4A-PEtOz) [60], and eight-arm PiPrOx with calix [8] arene core (C8A-PiPrOx) [69]. For six- and four-arm PEtOz, the considered dependences differ insignificantly, and only in the region of high NaCl content, the Δ*T*_1_ value for CPh6-PEtOz is noticeably higher than Δ*T*_1_ for C4A-PEtOz. For star-shaped PiPrOx at all concentrations, the Δ*T*_1_ value for the polymer with CPh6 core is higher than for the star with C8A. These facts suggest that star-shaped pseudo-polypeptoids with a hexaaza[2_6_]orthoparacyclophane core are more sensitive to the presence of NaCl than similar stars with a calix[n]arene core. However, it should be remembered that the compared polymers differed not only in the structure of the branching center, but also in the number and length of arms. The values of the latter characteristics determine the intramolecular density of the macromolecule, and, accordingly, the accessibility of the core for solvent molecules and molecules of low molecular weight salts.

## 4. Conclusions

The effect of NaCl and *N*-PTS on the self-organization in aqueous solutions of six-arm star-shaped CPh6-PAlOz and CPh6-PAlOx and on phase separation temperatures were investigated. It was shown that in the case of CPh6-PAlOx at low temperatures, the addition of salts does not lead to significant changes in the solution characteristics. A different situation took place for CPh6-PAlOz, in solutions of which, with a salt content corresponding to approximately one salt molecule per arm of star, the set of scattering objects changed. At this concentration, the micelle-like structures appeared in solutions, and isolated molecules ceased to be detected by dynamic light scattering. The observed effect depended on the arm structure. In CPh6-PEtOz solutions, micelle-like aggregates appeared with the addition of both salts, while in CPh6-PiPrOz solutions they formed only with *N*-PTS addition. In NaCl solutions of CPh6-PiPrOz, macromolecules and large aggregates were present in solutions at all studied salt concentrations. The effect of the salt structure was traced in the fact that in most N-PTS solutions the sizes of the aggregates were constant, while in NaCl solutions they increased with growth of salt concentration.

On heating, a phase transition with the formation of supramolecular micron-sized structures was observed in all the studied water–salt solutions of the star-shaped CPh6-PAlOz and CPh6-PAlOx. As well as in aqueous solutions, in both used solvents, at the same salt concentration, the phase separation temperature decreased in the series CPh6-PEtOx–CPh6-PEtOz–CPh6-PiPrOx–CPh6-PiPrOz. This is caused by an increase in the hydrophobicity of the polymers both with growth of the size of the side radical in the arms and with an elongation of the monomer unit by one –CH_2_– group.

The effect of the structure of salt and polymer on the phase separation temperature *T*_1_ was found. For all the stars studied, the temperature *T*_1_ monotonically decreased with increase in NaCl content in solution from *c*_salt_ = 0 to 0.154 M. This reduction for CPh6-PEtOz and CPh6-PEtOx polymers reached 49 and 37 °C, respectively. For more hydrophobic stars with isopropyl side groups, the discussed change was much smaller, 23 °C for CPh6-PiPrOx and 11 °C for CPh6-PiPrOz. In *N*-PTS solutions for all polymers, the dependence of the phase separation temperature on the salt concentration was non-monotonic. In the region of low salt content, *T*_1_ decreased sharply, reaching a minimum value at concentration *c*_salt_ corresponding to approximately one *N*-PTS molecule per one arm of a polymer star. Above this concentration, an increase in the phase separation temperature was observed. As well as in NaCl solutions, in solutions with the addition of *N*-PTS, the maximum change in *T*_1_ was greater for polymers with ethyl side radicals. Comparison of the obtained results with the literature data for star-shaped pseudo-polypeptoids with a calix[n]arene branching center showed that PAlOz and PAlOx stars with a hexaaza[2_6_]orthoparacyclophane core are more sensitive to the presence of salt in solution.

## Figures and Tables

**Figure 1 polymers-13-01152-f001:**
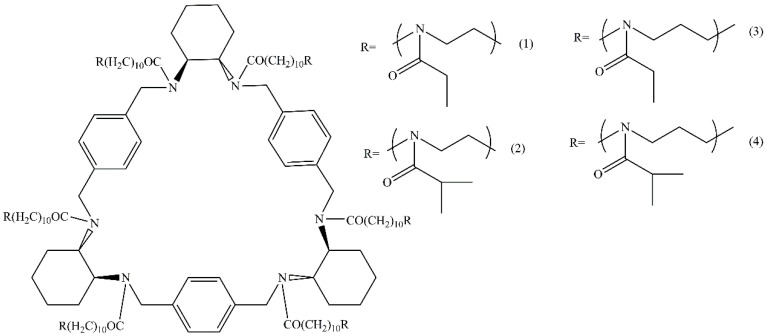
Structure of star-shaped poly-2-ethyl-2-oxazoline (**1**), poly-2-isopropyl-2-oxazoline (**2**), poly-2-ethyl-2-oxazine (**3**), and poly-2-isopropyl-2-oxazine (**4**).

**Figure 2 polymers-13-01152-f002:**
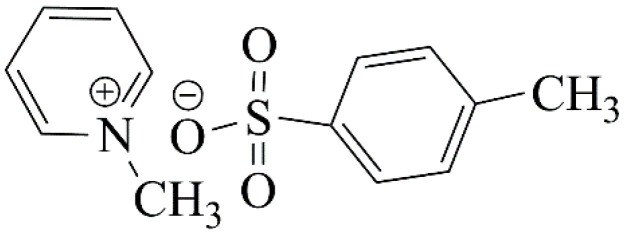
Structure of *N*-methylpyridinium *p*-toluenesulfonate.

**Figure 3 polymers-13-01152-f003:**
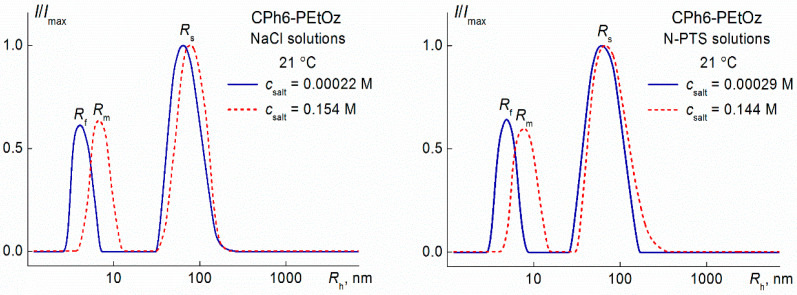
Dependences of relative light scattering intensity *I*/*I*_max_ on the hydrodynamic radius *R*_h_ of scattering species for CPh6-PEtOz solutions at 21 °C. (Variables *R*_f_, *R*_m_ and *R*_s_ will be discussed in Section 3.1).

**Figure 4 polymers-13-01152-f004:**
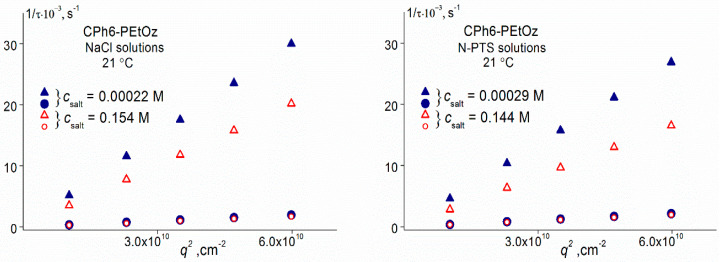
Relaxation time 1/τ on squared wave vector q^2^ for CPh6-PEtOz solutions at 21 °C.

**Figure 5 polymers-13-01152-f005:**
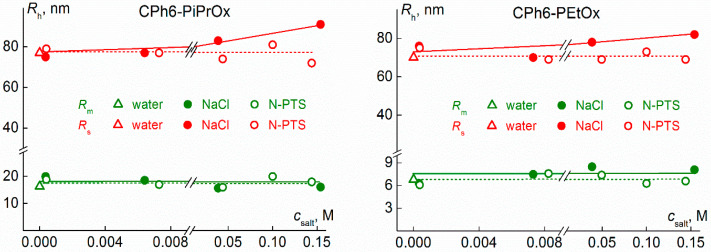
Dependences of hydrodynamic radii *R*_m_ and *R*_s_ of scattering objects on salt concentration *c*_salt_ for solutions of CPh6-PiPrOx at 11 °C and CPh6-PEtOx at 21 °C. In the Figure 5, Figure 6, Figure 11 and Figure 12 for all polymers and for both salts, the first point on the dependences refers to an aqueous solution, and the second, third and fourth points correspond to the following salt contents: one salt molecule per one macromolecule, per one arm of the polymer star, and per one monomer unit, respectively.

**Figure 6 polymers-13-01152-f006:**
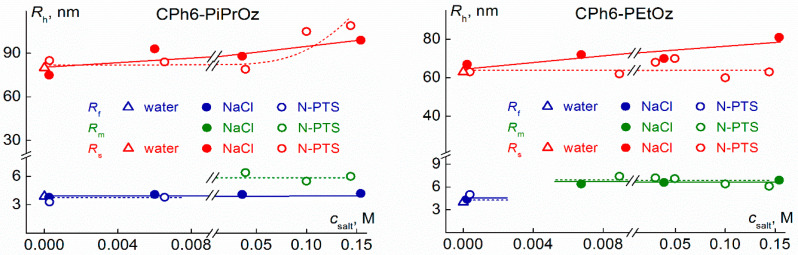
The dependences of the hydrodynamic radii *R*_f_, *R*_m_ and *R*_s_ on the salt concentration *c*_salt_ for solutions of CPh6-PiPrOz at 11 °C and CPh6-PEtOz at 21 °C.

**Figure 7 polymers-13-01152-f007:**
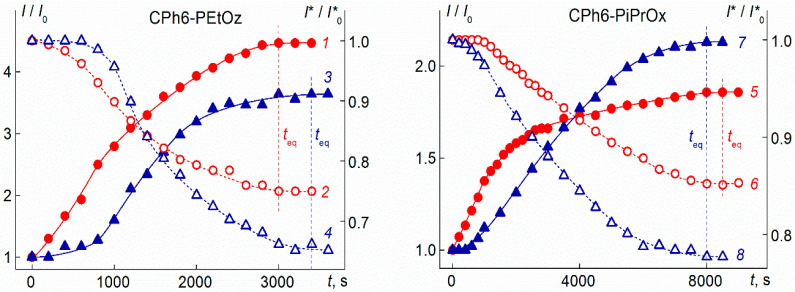
Dependences of relative light scattering intensity *I*/*I*_0_ (1, 3, 5, and 7; closed symbols) and optical transmission *I**/*I**_0_ (2, 4, 6, and 8; open symbols) on time t for water–salt solutions of investigate polymer stars. Left panel: CPh6-PEtOz solutions at NaCl concentration *c*_salt_ = 0.00674 M and *T* = 62 °C (1 and 2) and at *N*-PTS concentration *c*_salt_ = 0.00892 M and *T* = 31 °C (3 and 4). Right panel: CPh6-PiPrOx solutions at NaCl concentration *c*_salt_ = 0.00641 M and *T* = 38 °C (5 and 6) and at N-PTS concentration *c*_salt_ = 0.00803 M and *T* = 27 °C (7 and 8). Vertical lines mark the *t*_eq_ value. *I*_0_ and *I**_0_ are values of light scattering intensity and optical transmission at *t* = 0, respectively. In this Figure 7 and in Figure 8 and Figure 9, the salt concentrations correspond to one salt molecule per one arm of the polymer star.

**Figure 8 polymers-13-01152-f008:**
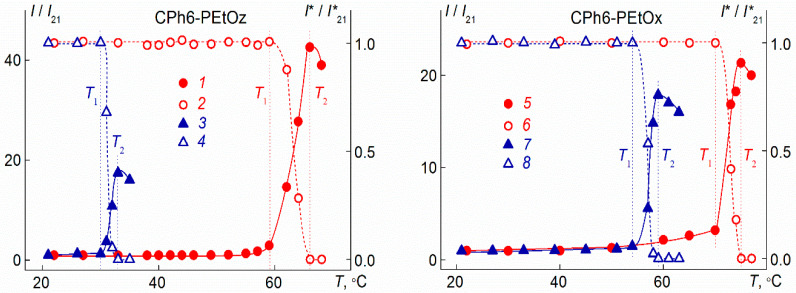
Dependences of relative light scattering intensity *I*/*I*_21_ (1, 3, 5, and 7; closed symbols) and optical transmission *I**/*I**_21_ (2, 4, 6, and 8; open symbols) on temperature *T* for water–salt solutions of investigated polymer stars. Left panel: CPh6-PEtOz solutions at NaCl concentration *c*_salt_ = 0.00674 M (1 and 2) and at *N*-PTS concentration *c*_salt_ = 0.00892 M (3 and 4). Right panel: CPh6-PEtOx solutions at NaCl concentration *c*_salt_ = 0.00730 M (5 and 6) and at *N*-PTS concentration *c*_salt_ = 0.00824 M (7 and 8). The vertical lines indicate the temperatures of the onset *T*_1_ and the end of the *T*_2_ phase transition for a given solution. *I*_21_ and *I**_21_ are values of light scattering intensity and optical transmission at 21 °C, respectively.

**Figure 9 polymers-13-01152-f009:**
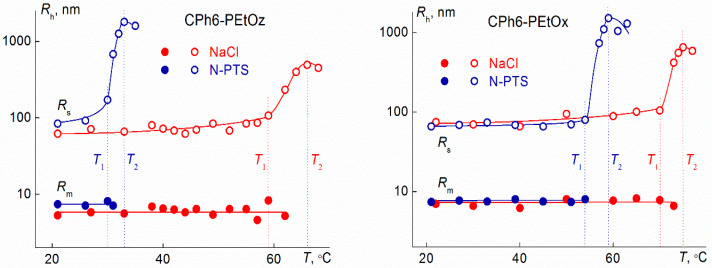
Dependences of hydrodynamic radii *R*_m_ (closed symbols) and *R*_s_ (closed symbols) on temperature *T* for CPh6-PEtOz solutions at NaCl concentration *c*_salt_ = 0.00674 M and at *N*-PTS concentration *c*_salt_ = 0.00892 M and for CPh6-PEtOx solutions at NaCl concentration *c*_salt_ = 0.00730 M and at *N*-PTS concentration *c*_salt_ = 0.00824 M.

**Figure 10 polymers-13-01152-f010:**
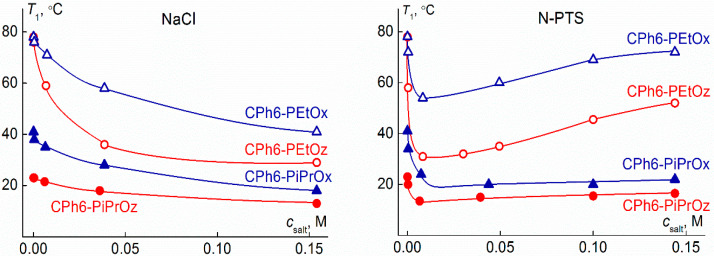
Dependencies of the phase separation temperature *T*_1_ on salt concentration *c*_salt_ for star-shaped CPh6-PAlOz and CPh6-PAlOx in NaCl and *N*-PTS solutions.

**Figure 11 polymers-13-01152-f011:**
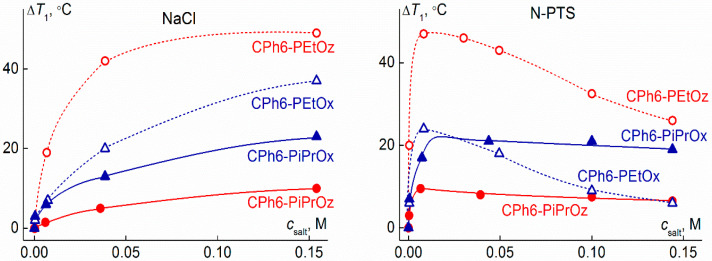
Dependencies of Δ*T*_1_ on salt concentration *c*_salt_ for star-shaped CPh6-PAlOz and CPh6-PAlOx in NaCl and *N*-PTS solutions.

**Figure 12 polymers-13-01152-f012:**
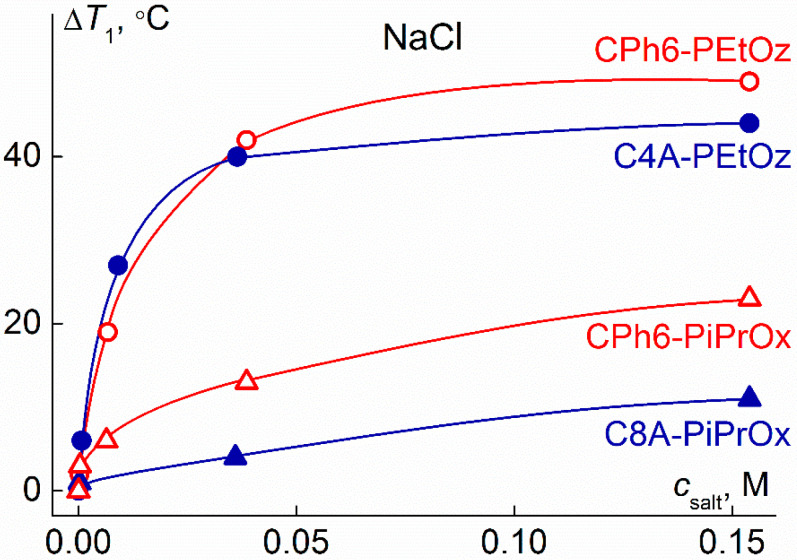
Dependencies Δ*T*_1_ on NaCl concentration *c*_salt_ for CPh6-PEtOz, CPh6-PiPrOx, C4A-PEtOz, and C8A-PiPrOx at *c* = 0.0050 g⋅cm^−2^.

**Table 1 polymers-13-01152-t001:** Values of <*t*_eq_^(max)^> for solutions of the studied polymer stars.

Polymer	*t*_eq_^(max)^, *s*	
	NaCl	*N*-PTS
CPh6-PEtOz	4500	3800
CPh6-PiPrOz	8800	8000
CPh6-PEtOx	3600	4200
CPh6-PiPrOx	7500	8500

## Data Availability

Not applicable.

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
