# Peer review of "Influence of Salt on the Self-Organization in Solutions of Star-Shaped Poly-2-alkyl-2-oxazoline and Poly-2-alkyl-2-oxazine on Heating"

_polymers, 2021, doi:10.3390/polym13071152_

Round 1

Reviewer 1 Report

The paper reports on a systematic study of the influence of salts on self organization of star shaped polyoxazoline and polyoxazine on heating.

Although the studied polymers are not new, their solution behavior, under certain conditions, is studied in detail.

The work is well designed and performed and can be accepted after some minor changes like:

  • Line 182: The concentration Csalt should be checked.
  • Line 286: the word “methylpipridinium” should be corrected.

Author Response

Dear reviewer,

thank you for your careful work with our article.

We corrected the text of the article in accordance with the comments of all reviewers.

  1. Line 182: The concentration Csalt should be checked.

            We have corrected the value of salt concentration.

  1. Line 286: the word “methylpipridinium” should be corrected.

            Taking in the mind the comments of reviewer # 3, we made significant changes to the this paragraph (lines 272 - 292).

Reviewer 2 Report

Please see the attached file below.

Author Response

Dear reviewer,

thank you for your careful work with our article.

In accordance with your comments and comments of the comments of other reviewers, we have corrected the manuscript.

1) I do not understand the motivation for using N-methylpyridinium p-toluenesulfonate (N-PTS) as a model for cetylpyridinium chloride (CPC). CPC and N-PTS have markedly different chemical structures for both the anion and cation, and so it is difficult to imagine their influence on the self-organization of the star-shaped polymers could be comparable.

            Indeed, the structures of N-methylpyridinium p-toluenesulfonate (N-PTS) as a model for cetylpyridinium chloride differ, but the main structural motif, namely, the presence of an alkylpyridinium fragment, is identical. In addition, the replacement of cetylpyridinium to methylpyridinium prevents the appearance of pyridinium salt micelles, which simplifies the interpretation of experimental data.

2) Thought the light scattering measurement is described in Reference #55, a short description would be useful in this manuscript. Especially the process for resolving and determining the multiple hydrodynamic radii for a single sample; a specific concentration of N-PTS in Figure 6 for CPh6-PiPrOz has three different radii, two of which are ~3 nm apart. Resolving the from light scattering data is non-trivial and the approach should be detailed.

            In accordance with your recommendation, we have corrected the description of the light scattering experiment. As for Figure 6, we found a typo (extra point) on it. Figure was corrected.

3) Why is the light scattering data for CPh6-PiPrOx not included in Figure 5?

            We have presented a graph for CPh6-PiPrOx. Accordingly, corrections have been made to the text describing Figure 5.

4) Line 193 – incorrect variable appears to be used for objects of 6.9 nm in size.

            In this fragment of the manuscript, small aggregates with a radius of about 7 nm are discussed, therefore the Rm variable is used.

5) Given the time dependence of the solution behavior of the star-shaped polymers shown in Figure 5, have the authors measured data with both increasing and decreasing temperature. There may be significant hysteresis that could be interesting.

            Unfortunately, there are hardware limitations that make it impossible to carry out a full heating-cooling cycle when using discrete temperature changes. Indeed, for the CPh6-PiPrOz and CPh6-PiPrOx solutions, the heating cycle was 15–20 hours. You can try to quickly heat the solution, and then cool it discretely. However, as shown by Winnik F., the heating rate can strongly affect the characteristics of aggregates formed in solutions. We carried out such an experiment and found that with fast and slow heating, the set of scattering objects in CPh6-PiPrOz and CPh6-PiPrOx solutions (at high temperatures) is different. Consequently, the cooling cycle will refer to a different physical state of the solution.

For some CPh6-PEtOz and CPh6-PEtOx solutions (with the shortest times for the “equilibrium” state establishment), we managed to go through a full heating-cooling cycle. Indeed, hysteresis was observed. In particular, the phase separation temperature decreased by 5 - 15 °C. At similar temperatures, the settling times during heating and cooling were the same within the experimental error. However, the data obtained are insufficient to draw any conclusions. Therefore, we believe that it is not correct to discuss this problem.

6) Figures 7, 8 and 9 – Which data (intensity of transmission) is represented by open or closed symbols should be made clear in the caption.

            The captions to figures have been corrected.

7) Concentrations of the two salts used are described on Page 4 (Lines 112 – 117) as being related to the number of star-shaped molecules, arms, etc, but then concentrations are given in mM for the data Figures. Unfortunately the molar concentrations of the, for example, one salt per macromolecule solution have not been clearly defined, so it is difficult to interpret the data. Figure 8, left panel, uses shows data for NaCl = 0.00674 M and N-PTS = 0.00892 M; do these concentrations represent one salt per macromolecule, arm or monomer unit for each respective salt.

            Thank you for your comment. We have made the necessary explanations in the captions to Figures.

8) Page 8, Lines 282 – 286 – the authors postulate N-PTS displays both kosmotropic and chaotropic behavior, and so study the influence of N-methylpyridinium iodide (N-PI). The behavior of N-PI is similar to N-PTS and so the authors conclude the constant N-methylpyridinium cation determines the salt behavior; this conclusion cannot be supported by the presented data. Without also examining a system with a constant anion it is not possible to conclude on the basis of two systems that N-methylpyridinium is controlling behvior. This study should be expanded or removed.

            We agree with your remark. Indeed, our conclusion is somewhat hasty. Since we cannot analyze the behavior of systems with a constant anion, we, on your recommendation, decided to remove the fragment under discussion. Accordingly, we have corrected Figure 10.

Reviewer 3 Report

The paper is devoted to the properties of water-salt solutions of star-shaped six-arm poly-2-alkyl-2-oxazines and poly-2-alkyl-2-oxazolines with the core of hexaaza[26]orthoparacyclophane. NaCl and N-methylpyridinium p-toluenesulfonate were used as salts.

The novelty of this paper is rather average. Therefore, I would recommend the publication of this paper in another journal. There are some ambiguities in the text.

The authors are requested to reconsider the following points before publication.
1) In section 3.1 there is no clear discussion of the solution behavior of star polymers.

Discussion of the behavior of star polymers CPh6-PAlOx (p.4, line 139, 143, 146) and CPh6-PEtOx (p.4, line 141-142, 145-146) does not agree with data shown in Figure 3. The respective figure for CPh6-PiPrOx should be shown.

2) In Table 1 the value of teq(max) for CPh6-PEtOz in NaCl solution should be 3000 (cf. Figure 5 (left panel)). The values of teq(max) for CPh6-PiPrOx should be 8800 in NaCl solution and 8000 in N-PTS solution (cf. Figure 5 (right panel)).

3) p.7, lines 253,254; it is reported that: for CPh6-PEtOx in NaCl solutions, a reliably measurable increase in I was observed at 45 °C,…

- according to line 5 in Figure 6 (right panel) - the real temperature is ~70°C

4) p.7, line 255; it is reported that: The change in Rs is not high,…

- it should be rather Rm instead Rs

Author Response

Dear reviewer,

thank you for your careful work with our article.

We corrected the text of the article in accordance with your comments and the comments of other reviewers.

1) In section 3.1 there is no clear discussion of the solution behavior of star polymers.

Discussion of the behavior of star polymers CPh6-PAlOx (p.4, line 139, 143, 146) and CPh6-PEtOx (p.4, line 141-142, 145-146) does not agree with data shown in Figure 5. The respective figure for CPh6-PiPrOx should be shown.

            We have presented a graph for CPh6-PiPrOx. Accordingly, corrections have been made to the text describing Figure 5.

            Figure 5 shows that

(i) at all salt concentrations in CPh6-PAlOx solutions, there are two types of species, the sizes of which are very different;

(ii) the radius Rm of small aggregates is independent of concentration of N-PTS and NaCl;

(iii) the radius Rs of large aggregates increases with increasing concentration of NaCl, but does not depend on the N-PTS content.

Comparison of the radii of the particles present in the aqueous-salt solutions of the studied CPh6-PAlOx with the hydrodynamic size of their molecules, which was determined in an organic solvent (see reference [55]), allows us to conclude that the discussed particles are aggregates. In addition, it makes it possible to roughly estimate the aggregation degree.

            The data for CPh6-PAlOz are discussed in a similar way, in solutions of which, depending on the salt content, either molecules and large aggregates or aggregates of different sizes were observed.

            Section 3.1 has been revised to make the text clearer.

2) In Table 1 the value of teq(max) for CPh6-PEtOz in NaCl solution should be 3000 (cf. Figure 7 (left panel)). The values of teq(max) for CPh6-PiPrOx should be 8800 in NaCl solution and 8000 in N-PTS solution (cf. Figure 7 (right panel)).

            Indeed, the values of establishment time teq in Figures 7 and the teq(max) value in Table 1 are not the same. This could be expected, since Figures 7 show the data for a given salt content and temperature, while Table 1 shows the average (under salt concentration) values teq(max), i.e. these are different parameters. We have corrected the manuscript, in particular the table header.

3) p.7, lines 253,254; it is reported that: for CPh6-PEtOx in NaCl solutions, a reliably measurable increase in I was observed at 45 °C, - according to line 5 in Figure 8 (right panel) - the real temperature is ~70°C.

            Changes in the light scattering intensity I with increasing temperature begin at relatively low temperatures. The error in determining I does not exceed 3%, therefore, if the I value has changed by 5-6%, it can be assumed that an increase in the light scattering intensity has begun.

            For the sample under discussion at 45 °C, the I value exceeds I at 21 °C (I21) by 10%, i.e. I/I21 = 1.1. With further heating of the solution, the light scattering intensity slowly increases, and at 70 °C I/I21 = 3.3. Temperature 70 °C is the temperature of onset of phase separation T1 for this solution (turbidimetry data). Above T1, the rate of change in the light scattering intensity on heating increases by an order of magnitude.

            We have corrected the discussed fragment of the manuscript.

4) p.7, line 255; it is reported that: The change in Rs is not high,…- it should be rather Rm instead Rs

            This is really Rs.

            This fragment of the manuscript has been corrected.
